# Laser Ignition of Potassium Picrate with Multi-Walled Carbon Nanotube Additives

**DOI:** 10.3390/molecules30040935

**Published:** 2025-02-18

**Authors:** Jianhua Wang, Jinjian Chen, Chen Shen, Yucun Liu, Junming Yuan, Yanwu Yu

**Affiliations:** School of Environment and Safety Engineering, North University of China, Taiyuan 030051, China; jinong1989@163.com (J.C.);

**Keywords:** potassium picrate (KP), multi-walled carbon nanotubes (MWNTs), laser ignite

## Abstract

Experimental investigations of the diode-laser-induced ignition of potassium picrate (KP) with a multi-walled carbon nanotube (MWNT) additive are presented in this article. KP/MWNT composites with varying contents were prepared directly by adding different quantities of MWNTs to a KP solution after the last synthesis step. Due to capillary action, the MWNTs homogeneously coated the surface of the KP, and some KP crystallized inside the MWNTs. The samples were characterized by scanning and transmission electron microscopy, differential thermal analysis, and laser ignition tests. At a constant laser power density, the doped KP showed a much shorter ignition delay time than the undoped KP (from 28.8 ms to 4.5 ms). Therefore, the higher the dopant MWNT ratio is, the shorter the ignition delay time is. Additionally, the more MWNTs are used to dope KP, the lower the required ignition power is.

## 1. Introduction

In recent decades, laser ignition has gained widespread attention in the field of energetic materials [1,2,3]. By utilizing high-energy laser beams as a heat source and precisely controlling the laser energy density, the target material can be heated to its ignition temperature, triggering combustion or detonation reactions. Compared with traditional initiation methods (e.g., electric sparks, hot wires, mechanical impact, and friction), laser ignition offers numerous advantages, including safety and reliability, long ignition distances, cost reduction, and environmental friendliness [4]. In the aerospace sector, laser ignition can efficiently ignite propellants, reducing the use of traditional ignition materials such as gunpowder and thereby enhancing safety and system reliability. Currently, most researchers improve the light sensitivity of energetic materials using light- or heat-absorbing doping agents, which lower the laser ignition energy [5,6,7]. For example, doping gold nanoparticles into RDX significantly enhances the laser ignition capability, reducing the laser ignition power density from over 914.4 W·cm^−2^ to below 79 W·cm^−2^ [8]. The performance of a laser during ignition is related to factors such as the laser type, energy power, and beam width; the material type; and the particle size [9,10]. A.V. et al. studied the critical conditions for the laser initiation of PETN with different metal additives using a microhotspot model [11]. Sun Mengxia et al. investigated changes in the ignition and combustion processes of AlH_3_ with different particle sizes and found that smaller particles have an improved combustion performance [12].

Potassium picrate (KP) is a thermally stable, heat-insensitive explosive commonly used as an initiator or a delay agent or added to detonation agents to accelerate the transition from deflagration to detonation [13,14,15]. As the primary explosive in hot-wire detonators, KP is sufficiently sensitive and releases sufficient energy [16]. KP can be used both as a standalone igniter and in combination with potassium perchlorate to accelerate ignition [17]. The formation and growth of the initial hotspot during the pulse excitation ignition of KP is relatively slow [18]. However, KP can also be mixed with explosives, such as HMX, HNS, and PETN, and be used in various detonators and delay elements. Currently, there is limited research on the laser ignition of KP, but potassium picrate-based compounds have significant potential as laser ignition agents [19].

Carbon nanotubes (CNTs), one-dimensional allotropes of carbon with a cylindrical hollow nanostructure, have emerged as promising light-sensitive agents due to their exceptional light absorption and thermal conductivity properties [20,21]. As thermal conductors, carbon-based nanostructures play a crucial role in transferring heat from the reaction zone to the unburned portion of the propellant [22]. Furthermore, the incorporation of CNTs into energetic materials can notably improve the ignition and combustion performance due to their extremely high thermal conductivity enhancing the combustion rate of solid propellants [23].

In this study, we employed crystal structure control technology to prepare KP materials with specific crystal structures and introduced multi-walled carbon nanotubes (MWNTs) with excellent thermal conductivity and light-sensitive properties. The aim of this research was to systematically investigate the thermal conduction and laser ignition properties of KP/MWNT composite materials by adjusting the MWNT content. We characterized the thermal decomposition and weight loss behavior of the composite materials and conducted laser ignition experiments to test the impact of different MWNT contents on the ignition performance of the KP materials. We explored their response characteristics and ignition delay time under laser radiation. These experiments were used to further assess the thermal behavior and ignition characteristics of the KP/MWNT composites under high-energy laser driving, providing theoretical and experimental data support for the design and optimization of future laser ignition materials.

## 2. Experimental Section

### 2.1. Preparation of KP/MWNT Composites

Multi-walled carbon nanotubes: tube diameter, 20~40 nm; wall thickness, 10~15 nm; inner diameter, 5~20 nm; number of walls, approximately 30~50 layers; length, less than 2 mm; purity, greater than 97%; specific surface area, SSA: >110 m^2^/g; tap density, 0.28 g/cm^3^; true density, 2.1 g/cm^3^; and electrical conductivity, EC: >100 S/cm. Picric acid (technical grade), magnesium oxide (analytical grade), and deionized water were prepared in the laboratory.

The synthetic process is shown in Figure 1. To obtain the Mg(PA)_2_ solution, 18.6 g of picric acid (HPA) and 2.43 g of magnesium oxide (MgO) were weighed in three beakers, and then 200 mL of deionized water was added. Each beaker was stirred and heated in a water bath at 60 °C until a brown solution formed. Each Mg(PA)_2_ solution was filtered, and 2 mL of a crystalline controlling agent was added. Then, 47 mL of KNO_3_ (8%) was dropped slowly into Mg(PA)_2_ while being stirred at 700 r·min^−1^. Finally, after the complete addition of KNO_3_, MWNTs with a mass fraction of 1 wt% were added to one of the beakers, and the resulting sample was labeled KP/MWNTs-a. The procedure was then repeated with MWNT mass fractions of 3 wt% and 5 wt%, and the corresponding samples were labeled KP/MWNTs-b and KP/MWNTs-c, respectively. Then, the mixture was moved into a low-temperature ultrasonic bath and stirred for 30 min in order for the particles to be uniformly distributed. The product was obtained by suction filtration, washed with deionized water, and dried at 50 °C.

### 2.2. Material Characterization

The crystal morphology of KP was characterized by scanning electron microscopy (SEM; EM-30PLUS; Beijing Tianyao Technology Co., Ltd., Beijing, China) operated at 20 kV. The physical structures of KP on the MWNT walls were observed with a transmission electron microscope (TEM; JEOL 2100F; JEOL (BEIJING) Co., Ltd., Beijing, China) operated at 200 kV. The KP crystal that covered the MWNT surface was washed until clean with hot water before the TEM test. The thermal properties of the KP/MWNT composites were analyzed using thermal gravimetric and differential thermal analysis (TG-DTA; HCT-1; Beijing Hengjiu Laboratory Equipment Co., Ltd., Beijing, China) with a heating rate of 10 °C·min^−1^ at temperatures ranging from 50 to 400 °C under N_2_ flow. The X-ray diffraction (XRD; Rigaku SmartLab SE, Tokyo, Japan) measurements were conducted at a scanning rate of 2°/min. Fourier-transform infrared (FTIR; Spectrum 100 Optica FT-IR Spectrometer; PerkinElmer, Shelton, CT, USA) spectroscopy was performed within a scanning range of 4000–600 cm^−1^ and a resolution of 1 cm^−1^.

### 2.3. Laser Ignition Test Method

The experimental setup used for laser ignition is depicted in Figure 1. The sample holder was a cylindrical stainless steel chamber with a Φ3 × 3 hole. About 10 mb of each sample was lightly pressed into the hole (filling density about 0.8 g·cm^−3^). The initiation source was a diode laser(Hamamatsu S1149; Hamamatsu Photonics K.K., Shizuoka, Japan) of 1.4 W with a wavelength of 450 nm in continuous wave mode. The size of the laser spot was adjusted using a lens with a 50 mm diameter and a 50 mm focal length. When the laser acted on the sample’s surface, the sample began to react, generating a flame. At the same time, the light signals of both the laser and the flame were collected using a photodiode and recorded with a digital oscilloscope (Agilent Technologies DSO7034B, Agilent Technologies, Inc., Beijing, China). Moreover, the photodiode with a filter merely detected the flame signal due to it filtering out the laser. The time difference between the two signals was measured as the ignition delay time. Each experiment was conducted in a dark environment, and the results were taken as the average from ten tests at each laser power.

## 3. Results and Discussion

### 3.1. Crystal Morphology

A visible color difference between KP samples was observable with the naked eye, with the pure KP crystal being yellow and the color changing to dark green because of MWNT doping. For more details, please see Figure 2. This means that the optical absorptivity greatly improved, which is in accord with other test results. The KP and KP/MWNT composites’ morphologies are shown in Figure 3. As we can see, the MWNTs were evenly spaced on the surface of the KP. On the other hand, due to capillary action, some aqueous solution infiltrated into the carbon nanotube walls during the KP/MWNT sample preparation. When the solvent evaporated out, some KP crystals crystallized in the carbon nanotube walls. As clearly displayed in Figure 4, there were some nano-crystals in the corner and the port of the nanotube wall. The particle size was about 20 nm. Since the concentration of the KP solution entering the MWNT tube walls was consistent, the crystal content in the tube walls of the three samples was essentially the same.

### 3.2. Thermal Analysis

The thermal properties of samples with different MWNT contents were tested using TG-DTA under a N_2_ flow. Figure 5 shows that the decomposition peak slightly advanced as the MWNT content increased. The pure KP exothermal peak was at 347.4 °C, and those of KP/MWNTs-a, KP/MWNTs-b, and KP/MWNTs-c were at 348.0, 345.3, and 344.3 °C, respectively. The MWNTs accelerated the KP thermal decomposition reaction because of their strong thermal conductivity. However, a small amount of individual MWNTs had only a minor effect on the thermal properties of KP, as the sample’s exothermal peak only slightly advanced (by less than 5 °C).

### 3.3. XRD and FTIR Analysis

The KP and MWNTs in their pure forms and as composite materials were analyzed in detail using XRD and FTIR, with the relevant curves shown in Figure 6. The X-ray diffraction (XRD) analysis revealed the unique crystal structure characteristic peaks of both KP and MWNTs. As the two materials were mixed, these characteristic peaks remained clearly visible in the XRD pattern of the mixture, without disappearing or undergoing significant shifts, indicating that the two materials maintained their respective crystal structures after mixing, with no significant chemical changes or reactions, and their molecular structures were preserved. The Fourier-transform infrared spectroscopy (FTIR) analysis further supported this conclusion. Both KP and MWNTs exhibited specific absorption peaks, reflecting the characteristic functional groups in their molecular structures. In the FTIR spectrum of the mixture, these characteristic absorption peaks were also clearly identifiable. This indicates that, although there may be interactions between the two materials during mixing, their individual components remain stable at both the physical structure and molecular levels, providing important experimental evidence for further research into their properties.

### 3.4. Laser Ignition

Tests were conducted in the laser power density range of 5.7~278 W·cm^−2^. The laser ignition delay times of KP with different MWNT contents are shown in Table 1. The results suggest that the ignition delay time decreased with increasing laser energy. When the laser beam irradiated the samples’ surfaces, the KP crystals absorbed thermal energy and started to decompose. At the same time, MWNTs, as strong optical absorbers and thermal conductors, transmitted this energy to the KP, accelerating the decomposition reaction. According to the heat transfer theory for laser ignition [24], the explosive material temperature increase is directly proportional to the laser power density and the square root of the delay time, as shown in Equation (1):(1)$$\Delta T=\frac{2l_{0}}{k}\sqrt{\frac{Kt}{\pi }}$$
where *l*_0_ is the laser power density, *k* is the thermal conductivity of the explosive material, *K* is the thermal diffusivity, and *t* is the ignition delay. Therefore, the higher the power density is, the less time it takes to reach the explosive ignition temperature. However, this equation is an ideal situation. In practice, thermal decomposition from the reaction zone must be taken into account. As we all know, the ignition reaction occurs when the ignition temperature of the energetic material is reached. As the laser power density increases, the inert heating time is shortened, and the temperature rises faster, so thermal decomposition in the reaction zone is accelerated and ignition occurs quickly. When the laser power density is further increased, the heating rate slows down because thermal decomposition in the reaction zone reaches a saturated state, so the ignition delay time no longer significantly changes.

These results also show that the ignition delay decreased as the MWNT content increased in the KP/MWNT composites. As shown in Table 1, when the laser energy density was 278 W·cm^−2^, the ignition delay time of KP was significantly reduced from 28.8 ms (pure KP) to 8.7 ms (KP/MWNTs-a, 1%). Those of KP/MWNTs-b (3%) and KP/MWNTs-c (5%) were both reduced to 4.5 ms, which indicates that the acceleration effect of MWNTs at these concentrations reached saturation under intense laser irradiation, and adding more MWNTs could not further reduce the ignition time. However, when the laser intensity was weakened, the advantage of more MWNTs was obvious (see Table 1). In addition, the minimum ignition power density decreased from 18 W·cm^−2^ (pure KP, 0%) to 7.8 W·cm^−2^ (KP/MWNT, 5%) as the MWNT content increased. The higher the MWNT content, the higher the optical absorbency and thermal conductivity efficiency values, meaning that less laser power was needed. All these results suggest that MWNTs are remarkably optically sensitive materials that can improve the laser ignition properties of KP.

The samples flashed and quickly burned down after laser exposure. To analyze the ignition process of the KP/MWNTs, pictures of the ignition behavior were taken at 420 frames per second. The progress of sample ignition and combustion is shown in Figure 7. We defined the moment that the laser beam contacted the sample surface as *t* = 0 ms and identified the ignition, growth, and degeneration stages. The burning duration of the KP/MWNT composites was variable: pure KP burned for about 80 ms; KP/MWNTs-c burned for about 230 ms; and the durations for KP/MWNTs-a and KP/MWNTs-b were 110 ms and 170 ms, respectively. Two possibilities can account for this phenomenon. First, the MWNTs, as inert materials, prevented the spread of chemical reactions between KP particles. Second, the MWNTs participated in a secondary reaction with the surrounding air, resulting in an increased flame duration, which can be proved by the spark shown in Figure 7b.

## 4. Conclusions

In this study, MWNTs, as optical sensitizers, were added to KP to improve its laser ignition properties. The MWNTs were directly added to a KP solution after the last synthesis step to produce a homogeneous coating on the KP surface. This approach to making KP/MWNT samples is simple and efficient. The KP crystals and the MWNTs were not simply mixed; due to capillary action, some nano-crystals became crystallized in the MWNT walls. Due to the strong thermal conductivity of MWNTs, the thermal decomposition reaction of KP slightly accelerated; the exothermal peak advanced by about 4 °C. The ignition delay time notably decreased with increasing MWNT content in the KP/MWNT composites. This rule was also true for the minimum ignition power density. All these results suggest that MWNTs are helpful optical sensitizers for the laser ignition of KP. At a power density of 278 W·cm^−2^, using even the smallest proportion of MWNTs as the dopant significantly reduced the ignition delay time, showing a similar performance to the sample with the maximum doping ratio at 140 W·cm^−2^. Further increasing the doping ratio resulted in minimal improvements in ignition performance.

## Data Availability

Data are contained within the article.

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
