# Peer review of "Laser Ignition of Potassium Picrate with Multi-Walled Carbon Nanotube Additives"

_molecules, 2025, doi:10.3390/molecules30040935_

Round 1

Reviewer 1 Report

Comments and Suggestions for Authors

Report on the submitted article Molecules - 3438479       

In this work the authors deal with the laser ignition of potassium picrate (KP) with MWNTs additives. They basically measure the ignition delay time and the minimum required laser density power for ignition as a function of the dopant ratio of MWNTs in the KP samples. The results are interesting showing a decrease of first the ignition time and second the minimum required energy for ignition versus doping. However, in my opinion the authors do not present clearly and explicitly their results and conclusions. This is mainly due to the poor English of the text. For example, in some sentences the verb is missing, while in some others the grammar is wrong. Therefore the manuscript needs a widespread language improvement. Based on that, I can’t suggest the publication of the paper in the present form. If the authors want to resubmit the paper they must drastically improve the manuscript. To this direction the following lists of remarks may help.

Majors

1) Introduction: The authors should explain better the motivation of this work, the reasons why they deal with this material (KP) in combination with MWNTs, what they expect and what they have achieved.

2) Fig.2: The authors made three different KP/MWNT samples (1,2 and 5% per weight). From what sample of KP/MWNT this image has been taken? It is unfortunate that the scale bar is not shown clearly. The image shows only a single orthogonal block of KP decorated on its surface with hairlooking MWNTs on the surface. Is this picture similar with the rest of the sample and moreover for all the rest samples?

3) Fig. 3: What sample this image represent?

3) Fig. 4: The y axis does not have legend.

4) How did the authors measure the ignition delay time? This is a relatively experimentaly short time. What is the acuuracy? These information should be mentioned in the experimental part.

5) Abstract: The authors should present more clearly and explicitly the basic conclusions of the paper. For example: “By keeping the laser power density constant on the samples, the doped KP showed much shorter ignition delay time than the undoped KP (28.8 ms to 4.5 ms). The higher dopant MWNTs’ ratio is, the shorter ignition delay time is recorded. Also the heavier MWNTs doping of KP is, the smaller the required ignition power.”

Minors

1) Reference 3: Please erase 1 after the names.

2) Introduction: Please explain the abbreviations HMX, HNS, RDX and PETN.

3) Caption of Scheme 1: Is this synthesis of KPA or KP?

4) Line 52: “…into a piece beaker…”. The whole sentence needs review.

5) Lines 105-106: “MWNTs have accelerated the KP thermal decomposition… bits of MWNTs have less effect…”.

6) Ref. 21: I think the right reference is “Östmark H., Laser as a Tool in Sensitivity Testing of Explosives. 8th Int. Symposium on Detonation, New Mexico, USA, July 15-19, 1985, p. 473-484.”

7) Lines 126-129: “As we see in the figure 5, the ignition delay time sharply decreases with the increasing of laser energy, when the laser power density is less than 50 W·cm-2. However, but when the laser power density becomes more than 50 W·cm-2, the ignition delay time smoothly decreases with the increasing of laser power.

8) Lines 137-138: “…with the increasing of MWNTs ratio in the KP/MWNTs composites.”

Comments on the Quality of English Language

The quality of English language is low.

Reviewer 2 Report

Comments and Suggestions for Authors

Manuscript ID: molecules-3438479

Title: Laser Ignition of Potassium Picrate with MWNTs Additives

Authors: Jianhua Wang et al.

Introduction. The authors wrote the introduction section unsatisfactorily. They did not provide a detailed description of the processes. In lines 23, 32, 36, the authors cite a huge number of articles, although according to the rules of the MDPI it is necessary to make no more than 3 references to one fact. The authors should completely rewrite this section and separately indicate the relevance and novelty of the research.

Section 3.1. Authors should add description for SEM and TEM images what sample they used: KP/MWNTs-a, KP/MWNTs-b or KP/MWNTs-c? It is necessary to write about difference of these samples using SEM and TEM, so authors should add 3 figures for SEM and 3 for TEM.

Figure 4. The quality of the figure is extremely low. Improve it to 300 DPI or higher. Why didn't the authors attach the TG curve, only the DTA?

Figure 5. Add the error bars for each point.

Figure 6. Authors should add images of the ignition and combustion process for KP/MWNTs-a and KP/MWNTs-b.

Conclusions. The conclusions are general, it is necessary to add a detailed description of the key conclusion, that is, the numerical values of the best experiment, its parameters, etc.

Reviewer 3 Report

Comments and Suggestions for Authors

This article is comprehensive, logically organized, and contains valuable information on the laser ignition of potassium picrate (KP) with multi-walled carbon nanotubes (MWNTs) additives.

To improve the manuscript, the author should take the following considerations:

(1) What are the inner and outer diameters of the MWNTs? How many walls are present in the MWNTs? The authors should include all information in the “2.1. Preparation of KP/MWNTs composites” section. Moreover, the authors should provide high-resolution transmission electron microscope (HR-TEM) images of the MWNTs.

(2) The authors should provide and discuss the nitrogen adsorption-desorption isotherms, specifically pore volume and structure, of KP/MWNTs composites, and include all information in the “2.2. Materials characterization” section.

(3) The authors presented the ignition delay time of KP with different MWNTs content in Table 1. This manuscript does not contain much of an error analysis on ignition delay time (ms) performance which is highly required for readability purposes. The authors should place the standard deviations to improve the ignition delay time and ensure the reliability and readability of the present research. The authors should calculate the ignition delay time using the density functional theory (DFT) modeling data.

Comments on the Quality of English Language

Abstract: The experimental investigations of diode laser-induced ignition of potassium picrate (KP) with multi-walled carbon nanotubes (MWNTs) additive are presented in this article. The KP/MWNTs composites with varying content were prepared directly by adding a certain quality of MWNTs in the KP solution during the last synthetic process. Due to the capillarity action, the MWNTs were homogeneously coated on the surface of KP and some KP crystals crystallized in the MWNTs’ tubes. The samples were characterized by scanning electron microscopy, transmission electron microscope, differential thermal analysis, and laser ignition tests. Compared with the pure KP, the ignition delay times were significantly reduced from 28.8 ms to 4.5 ms. KP's minimum ignition power decreased with the MWNTs addition due to its strong optical absorbency and thermal conductivity properties.

Reviewer 4 Report

Comments and Suggestions for Authors

In this manuscript, the author reported the laser ignition of potassium picrate with MWNTs additive, which is a meaningful work. However, the whole manuscript was in a bad organizing and writing, which is not meet the requirements of this journal.

1. The characterization and content is not sufficient to support an Article.

2. The Introduction not provided sufficient background, more references about the modification of KP should cited and discussed.

3. The section of Materials was missed, the detail information about all materials should provided.

4. Line 46, what is Mg(PA)2?

5. Line 60, what is KPA?

6. Line 86-87, “Compared with the pure KP by the naked eye, the color of KP crystal changed from yellow to dark green because of MWNTs doped.” where is the picture?

7. In the SEM image, the KP is much bigger than MWNTs, but in the TEM image, the KP is much smaller than MWNTs, why?

8. The data in table 1 are the same as figure 5, one of them should delete.

Comments on the Quality of English Language

The English could be improved to more clearly express the research.

Round 2

Reviewer 1 Report

Comments and Suggestions for Authors

Some last comments:

1) Figure 2 : SEM images ...

2) Figure 3 : TEM images ...

3) Figure 5: Is T=347 or 3468 oC for KP/MWNTs-a?

Reviewer 2 Report

Comments and Suggestions for Authors

The authors corrected all the shortcomings and answered all the questions from the experts.

In this form, the article can be accepted.

Reviewer 3 Report

Comments and Suggestions for Authors

The manuscript was revised carefully and improved according to the reviewer’s suggestions. The scientific insights are expressed well in this revised submission. The current revision is recommended for publication in the Special Issue: Advanced Carbon Nanomaterials and Their Applications in the Molecules.

Reviewer 4 Report

Comments and Suggestions for Authors

Other characterizations such as FTIR, XRD, Raman should added.

Comments on the Quality of English Language

The English could be improved to more clearly express the research
